# Linear Ubiquitination Mediates EGFR-Induced NF-κB Pathway and Tumor Development

**DOI:** 10.3390/ijms222111875

**Published:** 2021-11-02

**Authors:** Fang Hua, Wenzhuo Hao, Lingyan Wang, Shitao Li

**Affiliations:** Department of Microbiology and Immunology, Tulane University, New Orleans, LA 70112, USA; fhua@tulane.edu (F.H.); whao2@tulane.edu (W.H.); lwang32@tulane.edu (L.W.)

**Keywords:** EGF, LUBAC, HOIP, PKP2, linear ubiquitin, NF-κB, tumorigenesis

## Abstract

Epidermal growth factor receptor (EGFR) is a receptor tyrosine kinase that instigates several signaling cascades, including the NF-κB signaling pathway, to induce cell differentiation and proliferation. Overexpression and mutations of EGFR are found in up to 30% of solid tumors and correlate with a poor prognosis. Although it is known that EGFR-mediated NF-κB activation is involved in tumor development, the signaling axis is not well elucidated. Here, we found that plakophilin 2 (PKP2) and the linear ubiquitin chain assembly complex (LUBAC) were required for EGFR-mediated NF-κB activation. Upon EGF stimulation, EGFR recruited PKP2 to the plasma membrane, and PKP2 bridged HOIP, the catalytic E3 ubiquitin ligase in the LUBAC, to the EGFR complex. The recruitment activated the LUBAC complex and the linear ubiquitination of NEMO, leading to IκB phosphorylation and subsequent NF-κB activation. Furthermore, EGF-induced linear ubiquitination was critical for tumor cell proliferation and tumor development. Knockout of HOIP impaired EGF-induced NF-κB activity and reduced cell proliferation. HOIP knockout also abrogated the growth of A431 epidermal xenograft tumors in nude mice by more than 70%. More importantly, the HOIP inhibitor, HOIPIN-8, inhibited EGFR-mediated NF-κB activation and cell proliferation of A431, MCF-7, and MDA-MB-231 cancer cells. Overall, our study reveals a novel linear ubiquitination signaling axis of EGFR and that perturbation of HOIP E3 ubiquitin ligase activity is potential targeted cancer therapy.

## 1. Introduction

Epidermal growth factor receptor (EGFR) (also known as ErbB-1) is a member of the ErbB family of plasma membrane receptor tyrosine kinases (RTKs), which also includes ErbB-2 (HER2), ErbB-3, and ErbB-4. Upon stimulation, EGFR undergoes dimerization, autophosphorylation, and activation [1]. Overexpression and mutations result in EGFR constitutive activation, which has been shown in up to 30% of solid tumors, generally correlated with a poor prognosis [2,3,4]. Constitutive EGFR activation promotes cell survival, proliferation, and invasiveness by activating PI3K/AKT, signal transducer and activator of transcription (STAT), and nuclear factor-κB (NF-κB) pathways [4]. NF-κB activities are seen in multiple solid tumors and provide oncogenic signals to cancer cells. However, the mechanism of EGFR-mediated NF-κB activation is not well elucidated.

NF-κB plays an important role in the immune system and participates in cell survival, differentiation, and proliferation [5]. Various stimuli, such as TNFα, IL-1, LPS, and TLR ligands, activate NF-κB signaling. These ligands engage with their cognate receptors and recruit distinct adaptors, which all recruit and activate the IKK kinase complex consisting of IKKα, IKKβ, and NEMO. The activated IKK complex phosphorylates IκBα, which leads to K48-linked polyubiquitination of IκBα and subsequent protein degradation. The degradation of IκBα results in the release of cytosolic sequestered NF-κB. NF-κB is further phosphorylated, forms homo- or hetero-dimer, and translocates to the nucleus. In the nucleus, NF-κB dimer binds genomic DNA and activates gene expression. Growth factors promote NF-κB activation through ERBB members, but the underlying mechanisms are only now starting to be understood [4].

Recently, the head to tail-linked linear ubiquitination was found to be critical for NF-κΒ-dependent inflammatory signaling and immune responses [6]. The linear ubiquitin chain is assembled by the linear ubiquitin chain assembly complex (LUBAC), a ubiquitin E3 ligase complex comprising HOIP (also known as RNF31), HOIL-1L (also known as RBCK1), and SHARPIN [7,8,9,10,11] together with the E2 conjugating enzyme UBE2L3 [12]. Linear polyubiquitin plays a critical role in regulating the canonical NF-κB pathway, such as TNFα, IL-1, CD40, and TLR signaling pathways [7,8,9,10,11,13,14]. Linear ubiquitin is also involved in NOD2-mediated inflammatory signaling [15] and genotoxic stress [16]. Whether linear ubiquitination is required for EGF-mediated NF-κB activation is unknown.

Plakophilin 2 (PKP2), along with PKP1, PKP3, and PKP4, form a subgroup of catenin protein family that is characterized by armadillo repeats. They are composed of a basic N-terminal head domain, followed by a series of armadillo repeats [17,18]. PKPs are scaffold proteins that are essential for the formation of desmosome and stabilization of cell junctions. Our previous study found that PKP2 limits influenza A virus infection by disrupting the viral polymerase complex in the nucleus [19]. Recently, it has been reported that PKP2 promotes tumor development by enhancing ligand-dependent and independent EGFR dimerization and activation [20]. However, whether PKP2 is involved in EGFR-mediated NF-κB activation is unknown.

PKP2 has been reported to be present in the HOIP protein complex by proteomics [21]. As PKP2 is a known binding partner of EGFR [20], we hypothesized that EGFR might activate NF-κB through PKP2 and HOIP-mediated linear ubiquitination. In this study, we examined this hypothesis and found that PKP2 interaction with EGFR was required for EGFR-induced NF-κB activation. Furthermore, PKP2 recruited HOIP to EGFR upon EGF stimulation, leading to linear ubiquitination and IKK activation. Knockout of HOIP impaired EGF-induced NF-κB activity and reduced cell proliferation. HOIP deficiency also inhibited the growth of A431 epidermal xenograft tumors in nude mice by more than 70%. Administration of the HOIP inhibitor HOIPIN-8 inhibited NF-κB activation, proliferation, and clonogenicity in three cancer cell lines. Collectively, our study reveals a critical role of linear ubiquitination in EGFR-mediated NF-κB signaling and that HOIP is a potential drug target for cancer therapy.

## 2. Results

### 2.1. PKP2 Is Required for EGFR-Induced NF-κB Activation

To examine the role PKP2 in the NF-κB signaling pathway, we transfected FLAG-tagged armadillo (Arm) repeat-containing proteins, PKP2, PKP4, or β-catenin (CTNNB1) with NF-κB reporter into HEK293 cells. PKP2, but not other Arm repeat-containing proteins, activated NF-κB reporter activity (Figure 1a). Furthermore, PKP2-induced NF-κB reporter activity was dose-dependent (Figure 1b). Next, we determined which domain of PKP2 was required for NF-κB activation (Figure 1c). As shown in Figure 1d, the N-terminal domain (aa 1-348) and the C-terminal Arm repeats (aa 349-837) alone barely activated NF-κB reporter activity, suggesting that both domains are required for PKP2-induced NF-κB activation.

PKP2 is reported to interact with EGFR [20]. We further dissected the domains for PKP2-EGFR interaction. Co-immunoprecipitation (co-IP) found that the N-terminal amino acid 1-109 of PKP2 (N109) was sufficient to bind EGFR (Figure 1e). We also made EGFR mutants to examine which domain was required for PKP2 interaction (Figure 1f). As shown in Figure 1g, the kinase domain of EGFR was responsible for the interaction with PKP2. Next, we examined the effects of PKP2 deficiency on EGF-induced NF-κB activity. EGF-induced NF-κB activation was observed in A431 cells, an epidermoid carcinoma cell line expressing high levels of EGFR [22]. Thus, we knocked out PKP2 in A431 cells by CRISPR. Then, we stimulated wild type and PKP2 knockout A431 cells with EGF. As shown in Figure 1h, EGF induced S32 phosphorylation of IκBα and S536 phosphorylation of p65 in wild type cells. In contrast, such phosphorylation was dramatically impaired in PKP2 knockout cells. Furthermore, the mRNA expression of NF-κB-stimulated genes, including IL-8, IL-6, CCDN1, and COX2, was dramatically reduced in PKP2 knockout cells (Figure 1i–l). Taken together, these data suggest that PKP2 is required for EGFR-induced NF-κB activation and NF-κB-regulated gene expression.

### 2.2. PKP2 Interacts with HOIP

To determine the role of PKP2 in the NF-κB signaling pathway, we searched protein interaction databases to seek PKP2 interacting proteins. One study showed that PKP2 was found in the HOIP protein complex by proteomics [21]. As HOIP is the E3 ubiquitin ligase in the LUBAC complex and critical for various NF-κB signaling pathways induced by TNFα, IL-1β, CD40L, and NOD2 ligand [23], we hypothesize that PKP2 transduces EGF signaling through the LUBAC to activate NF-κB. To test this hypothesis, we first validated the interaction between HOIP and PKP2 by co-IP. Different tagged PKP2 and HOIP were co-transfected into HEK293 cells. Co-IP confirmed the interaction between PKP2 and HOIP (Figure 2a). We then examined the subcellular localization of FLAG-tagged HOIP and HA-tagged PKP2 in the A431 cells. Immunofluorescence assay (IFA) displayed a co-localization between PKP2 and HOIP (Figure 2b).

To define the domain(s) responsible for PKP2 and HOIP association, we first dissected the PKP2 domains. Co-IP found that the N-terminal region (aa 1-109) of PKP2 was sufficient to bind to HOIP (Figure 2c,d). Similarly, we made a panel of HOIP deletion mutants (Figure 2e). These HOIP mutants were co-transfected with PKP2 into HEK293 cells. Co-IP showed that the N563 mutant but not the N349 interacted with PKP2, suggesting that the region of aa 349-563 was responsible for the interaction with PKP2 (Figure 2f). The aa 349-563 region consists of two Npl4 zinc finger (NZF) domains. Thus, we further made a construct expressing these two NZFs. Co-IP further found that the NZFs interacted with the N-terminal domain of PKP2 (Figure 2g). The combined experiments indicate that the NZFs of HOIP are necessary and sufficient for the binding to the N-terminus of PKP2.

### 2.3. PKP2 Activates NF-κB via HOIP

As PKP2 interacts with HOIP, we examined the effects of ectopic expression of PKP2 on HOIP- and LUBAC-induced NF-κB reporter activities. PKP2 was co-transfected with HOIP or LUBAC together with the NF-κB reporter into HEK293 cells. A low amount of HOIP and LUBAC was used to induce minimal reporter activity. PKP2 synergistically induced NF-κB reporter activities with HOIP and LUBAC (Figure 3a,b). Furthermore, the PKP2 mutant lacking the first 109 amino acids failed to activate NF-κB (Figure 3c), which is consistent with the requirement of first 109 amino acids for HOIP binding (Figure 2d). Next, we examined the effect of HOIP deficiency on PKP2 activity. As shown in Figure 3d, HOIP deficiency abolished PKP2-induced NF-κB reporter activity. By contrast, overexpression of HOIP was still able to activate NF-κB reporter in PKP2 knockout cells (Figure 3e), suggesting that HOIP is downstream of PKP2. Overall, our data demonstrate that PKP2 activates NF-κB via HOIP.

### 2.4. EGFR Activates Linear Ubiquitination and NF-κB via HOIP

As EGFR recruits PKP2 and PKP2 interacts with HOIP, we suspect that EGFR recruits HOIP via PKP2. We first examined the endogenous EGFR complex. PKP2 wild type and knockout cells were stimulated with EGF, then the EGFR complex was isolated using an anti-EGFR antibody. As shown in Figure 4a, EGF stimulation led to the recruitment of HOIP to EGFR in wild type cells. However, EGFR failed to recruit HOIP in PKP2 knockout cells upon EGF stimulation (Figure 4a). Furthermore, we transfected EGFR and HOIP with or without PKP2 into HEK293 cells. Co-IP found that PKP2 enhanced the interaction between EGFR and HOIP (Figure 4b). These data suggest that EGFR recruits HOIP via PKP2. Next, we examined NEMO linear ubiquitination, a hallmark for IKK activation mediated by LUBAC [8]. Significantly, EGF induced NEMO linear ubiquitination in A431 wild type cells, which is a critical sign of the activation of NF-κB signaling. However, linear ubiquitination was impaired in the HOIP knockout cells (Figure 4c). To determine whether HOIP is required for EGF-induced NF-κB activation, we stimulated wild type and HOIP knockout A431 cells with EGF. EGF induced S32 phosphorylation of IκBα and S536 phosphorylation of p65 in wild type cells (Figure 4d). By contrast, the phosphorylation of IκBα and p65 was dramatically impaired in HOIP knockout cells (Figure 4d). Furthermore, the mRNA expression of IL-8, IL-6, CCDN1, and COX2 was impaired in HOIP knockout cells (Figure 4e–h). Taken together, our data suggest that EGFR activates the linear ubiquitination of NEMO and NF-κB transcriptional activity via HOIP.

### 2.5. Deficiency of HOIP Suppresses A431 Cell Proliferation and Tumor Development

Because NF-κB activation is involved in cell proliferation, we conducted experiments to examine the role of HOIP in cell growth. The proliferation activity and clonogenicity were reduced in HOIP knockout A431 cells (Figure 5a,b). It was previously established that constitutive EGFR and NF-κB activities play a critical role in tumor development. To investigate the role of HOIP in EGFR signaling-mediated cancer development, we injected HOIP wild type or knockout A431 cells into the flanks of nude mice. Mice were monitored for tumor growth weekly and were sacrificed at 10 weeks after injection. HOIP knockout reduced the growth rate of tumors in nude mice by around 70% compared to the HOIP wild type tumors (Figure 5c,d). Furthermore, immunohistochemistry (IHC) demonstrated a much higher Ki-67 staining in the wild type tumor than HOIP knockout tumor (Figure 5e). Taking the findings together, we conclude that the depletion of HOIP inhibits A431 tumor cell proliferation and tumor development.

### 2.6. HOIP Inhibitor Suppresses A431 Cell Proliferation and Tumor Development

It has been reported that HOIPIN-8 inhibits HOIP E3 ligase activity and linear ubiquitination [23]. We treated A431 cells with the HOIP inhibitor, HOIPIN-8. As shown in Figure 6a, HOIPIN-8 inhibited IκBα and p65 phosphorylation. In line with the HOIP knockout data, HOIPIN-8 also inhibited mRNA expression of IL-8, IL-6, CCDN1, and COX2 induced by EGF in A431 cells (Figure 6b–e). Furthermore, the HOIP inhibitor HOIPIN-8 dramatically inhibited proliferation and clonogenicity of A431 tumor cells (Figure 6f,g). About 15% of A431 cells were subject to apoptosis (early plus late apoptosis) after HOIPIN-8 treatment (Figure 6h,i). More importantly, administration of HOIPIN-8 severely impaired the tumor growth capacity of A431 tumor and reduced the tumor size in nude mice (Figure 6j,k). Collectively, these data suggest that HOIPIN-8 inhibits A431 cell proliferation and tumor development.

### 2.7. HOIP Inhibitor Suppresses Breast Cancer Cell Proliferation and Clonogenicity by Blocking EGFR-Mediated NF-κB Activation

To extend our finding to other cancer cells, we treated two breast cancer cell lines, MCF-7 and MDA-MB-231, with HOIPIN-8. Similarly, HOIPIN-8 inhibited IκBα and p65 phosphorylation in MCF-7 and MDA-MB-231 (Figure 7a,b). Furthermore, HOIPIN-8 dramatically suppressed the proliferation of MCF-7 and MDA-MB-231 tumor cells (Figure 7c,d). Consistently, the HOIP inhibitor HOIPIN-8 also dramatically inhibited clonogenicity of MCF-7 and MDA-MB-231 tumor cells (Figure 7e,f). Taken together, these data suggest that HOIP inhibitor also inhibits EGFR-mediated NF-κB activation in breast cancer cells and suppresses breast cancer cell proliferation and clonogenicity.

## 3. Discussion

It is well known that EGF triggers NF-κB activation in various cell lines and tumors. Early studies demonstrate that EGF-induced NF-κB mainly comprises p65/p50 heterodimers [22]. Furthermore, EGF instigates NF-κB signaling by activating the IKK complex [24,25,26]. However, the signal cascade of the EGF-mediated NF-kB signaling pathway is still not well elucidated and is controversial in some cases. First, one study showed that Her2-induced NF-κB activation is dependent on PI3K/Akt, but not IKK [27]. By contrast, another study showed that Her2 activates NF-κB and induces invasion through the canonical pathway involving IKKα [28]. Second, in addition to conventional phosphorylation sites (serines 32 and 36) of IκBα, one study showed that EGF induces tyrosine phosphorylation on IκBα at tyrosine 42 [29]. However, it is unclear whether this phosphorylation is functionally relevant to EGF-induced NF-κB activation. Third, the EGF-NF-κB pathway is divergent upstream of the IKK complex. Several signal cascades have been reported to link EGF to the IKK complex, such as PKC-mediated phosphorylation of the CARMA3-Bcl10-MALT1 complex (CBM) [26], PLCγ-dependent and PKCε-mediated IKK phosphorylation [24], TNF receptor-associated protein 2 (TRAF2)- and TNF receptor-interacting protein (RIP)-mediated IKK activation [30,31], and the mTOR complex 2 pathway [32]. Despite the discrepancy and complexity, the literature largely agrees that EGF induces NF-κB transcriptional activity by activation of the IKK complex.

It is known that PKP2 associates with EGFR and facilitates EGFR-mediated tumorigenesis; however, how PKP2 promotes activation of EGFR is not well elucidated. It also needs to determine in which EGFR signaling axis PKP2 is involved and how PKP2 contributes to this signaling pathway. Here, we report the EGFR-PKP2-LUBAC signaling cascade as a bona fide signaling branch of EGFR-elicited NF-κB signaling pathway based on several lines of evidence. First, knockout of PKP2 and HOIP severely impairs IκBα and p65 phosphorylation, the hallmarks of NF-κB pathway activation. Deficiency of PKP2 and HOIP also reduces EGF-induced, NF-κB-regulated genes, such as IL-6, COX2, and CCND1. Furthermore, the HOIP inhibitor blocks EGF-induced IκBα and p65 phosphorylation and gene expression. Second, EGF induces linear ubiquitination of NEMO, the regulatory subunit of the IKK complex. Linear ubiquitination of NEMO has been shown to activate the IKK complex and downstream signaling. A previous study showed that NEMO prolonged tumor cell survival via regulation of apoptosis and activation of epithelial-to-mesenchymal transition, facilitating tumor metastasis [33]. Third, a previous study showed that EGF induces the recruitment of PKP2 to the EGFR complex [20]. Our study further demonstrates that EGFR also recruits and activates HOIP via PKP2. Last, HOIP knockout or HOIP inhibitor impairs tumor cell proliferation, clonogenicity, tumor growth, and development in the xenograft tumor mouse models. All these processes are fully or partially attributed to NF-κB activity. Taken together, we propose that EGFR recruits HOIP and its E3 complex LUBAC via PKP2 upon stimulation. Then, the LUBAC is activated to conjugate linear polyubiquitin onto the IKK complex, leading to IKK activation and subsequent NF-κB activation.

Currently, it is unclear how HOIP is activated by EGFR. HOIP activation is regulated by the OTU domain-containing deubiquitinase with linear linkage specificity (OTULIN). Recently, we found that TRIM32 conjugates non-proteolytic polyubiquitin onto OTULIN and the polyubiquitin blocks the interaction between HOIP and OTULIN, thereby activating NF-κB signaling [34]. OTULIN also interacts with HOIP via its PUB-interacting motif and inhibits LUBAC activity [35,36]. Phosphorylation of tyrosine 56 prevents OTULIN from binding to HOIP [35,36]. A recent study found that a non-receptor tyrosine kinase, ABL1, phosphorylates tyrosine 56 of OTULIN and promotes genotoxic Wnt/β-catenin activation [37]. However, ABL1-mediated phosphorylation of OTULIN has little effect on TNFα-induced NF-κB activation [37]. As EGFR is a tyrosine kinase and also recruits other tyrosine kinases, such as PI3K, upon stimulation, it is plausible that these tyrosine kinases phosphorylate tyrosine 56 of OTULIN, thereby activating HOIP and the LUBAC complex. We will further investigate this potential mechanism in the future by developing the tyrosine 56 phosphor-specific antibody of OTULIN.

The involvement of PKP2 and HOIP in the EGFR-NF-κB signaling implies the oncogenic role of both genes. PKP2 has been linked to cancer malignancy by several clinical and histological analyses [38,39,40,41,42,43]. PKP2 is reported to facilitate the proliferation and metastatic ability of the breast cancer cell line MDA-MB-231 and the development of tumors in xenograft mouse model [20]. Similarly, the role of LUBAC in cancers has emerged in recent years. LUBAC has functions in protecting cells from genotoxic damage-induced apoptosis and mediating NF-κB activation [16,44]. HOIP deficiency also causes embryonic lethality by aberrant TNFR1-mediated endothelial cell death [45]. These studies suggest that LUBAC might promote tumorigenesis and cancer development by preventing cell death. In line with this, a recent study shows that LUBAC accelerates B cell lymphomagenesis by conferring resistance to genotoxic stress on B cells [46]. Interestingly, HOIL-1L, another component of LUBAC, functions as the PKCζ ubiquitin ligase to promote lung tumor growth [47].

Furthermore, recent studies found that HOIP overexpression may contribute to cisplatin resistance in cancers. A recent study compared the expression profile data from a panel of several hundred cancer cell lines derived from tumors classified as cisplatin resistant or cisplatin sensitive. They found that cisplatin-resistant cancer cell lines show a significantly higher expression of HOIP and SHARPIN when compared with cisplatin-sensitive cancer cell lines [44]. Ruiz et al. showed that lung squamous cell carcinoma (LSCC) tumors were resistant to the chemotherapeutic agent cisplatin due to the higher expression of LUBAC and higher activity of NF-κB [48]. Furthermore, LUBAC inhibitors re-sensitized LSCC tumors to cisplatin [48]. Jo et al. showed that HOIP expression was elevated in activated B cell-like (ABC) diffuse large B cell lymphoma (DLBCL) and the LUBAC inhibitor suppressed the growth of lymphoma cells in the tumor transplantation model for human B cell lymphomas using a cell line derived from aCD19-cre-HOIP/MyD88LP mouse [46]. These data suggest that antagonizing LUBAC together with chemotherapy might be a potential therapeutic for chemotherapy-resistant tumors.

## 4. Materials and Methods

### 4.1. Mice

Female athymic Nude-Foxn1nu mice (20–25 g weight) were purchased from the Jackson Laboratory (Bar Harbor, ME, USA), and housed under specific pathogen-free (SPF) conditions with a controlled temperature of between 20–24 °C, humidity between 45–65%, and a 12 h light/dark cycle.

Wild type and HOIP knockout A431 cells were adjusted to a density of 5 × 10^7^ cells/mL in cold FBS-free DMEM, and then 100 µL cell suspension was mixed with 100 µL Matrigel. Following confirmation of successful inhalation anesthesia, 200 µL Matrigel-containing cells per mouse were transplanted subcutaneously into the flank region of 6–8-week-old female Nude-Foxn1nu mice. Tumor size measurements were initiated 7 days post-inoculation and monitored one time per week. Once tumors were palpable, dimensions of the tumor were measured externally using a caliper.

Animals inoculated with A431 cells were divided into the following groups. Group I received DMSO as a control and group II received HOIPIN-8 intratumorally. All the above drug treatments were started from day 14 following the tumor cell inoculation. This study was carried out in strict accordance with the recommendations in the Guide for the Care and Use of Laboratory Animals of the National Institutes of Health. The protocol was approved by the IACUC of Tulane University, New Orleans (protocol code 1003 and approval date 17 November 2020).

### 4.2. Cell Lines

HEK293, A549, A431, MCF-7, and MDA-MB-231 cells were obtained from the American Type Culture Corporation (ATCC). HEK293 were cultured in Dulbecco’s Modified Eagle Medium (Life Technologies, Carlsbad, CA, USA) containing antibiotics (Life Technologies) and 10% fetal bovine serum (Life Technologies) at 37 °C in a CO_2_ humidified chamber. A549, A431, MCF-7, and MDA-MB-231 cells (ATCC, Manassas, VA, USA) were cultured in RPMI Medium 1640 (Life Technologies) plus 10% fetal bovine serum and 1 × MEM Non-Essential Amino Acids Solution (Life Technologies).

### 4.3. Plasmids

Mutants of HOIP, EGFR, and PKP2 were constructed by PCR or using a Q5^®^ Site-Directed Mutagenesis Kit (New England Biolabs, Ipswich, MA, USA). Briefly, PCR was performed using FLAG-tagged human HOIP, EGFR, or PKP2 in pCMV3Tag-8 vector (Stratagene, Santa Clara, CA, USA) and the indicated primers (Table 1). After PCR, the PCR product is added to the Kinase-Ligase-DpnI (KLD) enzyme mix for circularization and template removal. The mixture was transformed into *E. coli* for generating plasmids. All mutants were validated by DNA sequencing. PKP2 mutants, N348 and C349, were reported previously [19]. The primers for cloning other PKP2 mutants, EGFR mutants, and HOIP mutants are listed in Table 1.

### 4.4. Antibodies

Primary antibodies were anti-FLAG (Sigma, St. Louis, MO, USA), anti-HA (Cell Signaling Technology, Boston, MA, USA), anti-GFP (Santa Cruz Biotechnology, Santa Cruz, CA, USA), anti-EGFR (Santa Cruz Biotechnology), anti-p65 (Cell Signaling Technology), anti-p-S536-p65 (Cell Signaling Technology), anti-p-S32-IκB (Cell Signaling Technology), anti-PKP2 (Fitzgerald Industries International, Concord, MA, USA), anti-HOIP (Bethyl Laboratories, Montgomery, TX, USA), anti-NEMO (Santa Cruz Biotechnology), anti-V5 (Thermo Scientific, Waltham, MA, USA), anti-β-actin (Abcam), anti-α-tubulin (Cell Signaling Technology), and anti-linear ubiquitin (Life Sensors, Philadelphia, PA, USA).

Secondary antibodies were Goat anti-Mouse IgG-HRP (Bethyl Laboratories, TX, USA), Goat anti-Rabbit IgG-HRP (Bethyl Laboratories), Alexa Fluor 594 Goat Anti-Mouse IgG (H + L) (Life Technologies), and Alexa Fluor 488 Goat Anti-Rabbit IgG (H + L) (Life Technologies, Carlsbad, CA, USA).

### 4.5. Sample Preparation, Western Blotting, and Immunoprecipitation

Approximately 1 × 10^6^ cells were lysed in 500 µL of tandem affinity purification (TAP) lysis buffer (50 mM Tris-HCl (pH 7.5), 10 mM MgCl_2_, 100 mM NaCl, 0.5% Nonidet P40, 10% glycerol, Complete EDTA-free protease inhibitor cocktail tablets (Roche, Basel, Switzerland) for 30 min at 4 °C. The lysates were then centrifuged for 30 min at 15,000× *g* rpm. Supernatants were collected and mixed with the Lane Marker Reducing Sample Buffer (Thermo Fisher Scientific, Waltham, MA, USA).

Western blotting and immunoprecipitation were performed as described in a previous study (38). Briefly, samples (10–15 μL) were loaded into Mini-Protean TGX Precast Gels, 15 wells (Bio-Rad, Hercules, CA, USA), and run in 1 × Tris/Glycine/SDS Buffer for 35 min at 200 V. Protein samples were transferred to Immun-Blot PVDF Membranes in 1 × Tris/Glycine Buffer at 70 V for 60 min. PVDF membranes were blocked in 1 × TBS buffer containing 5% Blotting-Grade Blocker (Bio-Rad) for 1 h. After washing with 1 × TBS buffer for 30 min, the membrane blot was incubated with the appropriately diluted primary antibody in antibody dilution buffer (1 × TBS, 5% BSA, 0.02% sodium azide) at 4 °C for 16 h. Then, the blot was washed three times with 1 × TBS (each time for 10 min) and incubated with secondary HRP-conjugated antibody in antibody dilution buffer (1:10,000 dilution) at room temperature for 1 h. After three washes with 1 × TBS (each time for 10 min), the blot was incubated with Clarity Western ECL Substrate (Bio-Rad) for 1–2 min. The membrane was removed from the substrates and then exposed to the Amersham imager 600 (GE Healthcare Life Sciences, Marlborough, MA, USA).

For immunoprecipitation, 2% of cell lysates were saved as an input control, and the remainder was incubated with 5–10 μL of the indicated antibody plus 20 μL of Pierce Protein A/G Plus Agarose (Thermo Fisher Scientific) or 10 μL of EZview Red Anti-FLAG M2 Affinity Gel (Sigma, St. Louis, MO, USA). After mixing end-over-end at 4 °C overnight, the beads were washed 3 times (5 min each wash) with 500 μL of lysis buffer. For ubiquitin detection, all beads were washed 3 times with 1 M urea for 15 min to exclude potential binding of unanchored polyubiquitin.

### 4.6. MTT Assays

Cells were seeded in a 96-well plate at a density of 400 cells/well in 100 μL culture medium. Then, cells were treated with DMSO or 30 μM HOIPIN-8. Cell growth rate was determined by the MTT Cell Proliferation Kit (Cayman Chemical, Ann Arbor, MI, USA).

### 4.7. Immunofluorescence Assay

Cells were cultured in the Lab-Tek II CC2 Chamber Slide System 4-well (Thermo Fisher Scientific). After the indicated treatment, the cells were fixed and permeabilized in cold methanol for 10 min at −20 °C. Then, the slides were washed with 1 × PBS for 10 min and blocked with Odyssey Blocking Buffer (LI-COR Biosciences, Lincoln, NE, USA) for 1 h. The slides were incubated in Odyssey Blocking Buffer with appropriately diluted primary antibodies at 4 °C for 16 h. After 3 washes (5 min per wash) with 1 × PBS, the cells were incubated with the corresponding Alexa Fluor conjugated secondary antibodies (Life Technologies, Carlsbad, CA, USA) for 1 h at room temperature. The slides were washed three times (5 min each time) with 1 × PBS and counterstained with 300 nM DAPI for 1 min, followed by washing with 1 × PBS for 1 min. After air-drying, the slides were sealed with Gold Seal Cover Glass (Electron Microscopy Sciences, Hatfield, PA, USA) using Fluoro-gel (Electron Microscopy Sciences).

### 4.8. Real-Time PCR

Total RNA was prepared using the RNeasy Mini Kit (Qiagen, Frankfurt, Germany). One µg quantity of RNA was reverse transcribed into cDNA using the QuantiTect reverse transcription kit (Qiagen). For one real-time reaction, 10 µL of SYBR Green PCR reaction mix (Eurogentec, Liege, Belgium) including 2 mg of the synthesized cDNA plus an appropriate oligonucleotide primer pair were analyzed on a 7500 Fast Real-time PCR System (Applied Biosystems, Waltham, MA, USA). The comparative Ct method was used to determine the relative mRNA expression of genes normalized by the housekeeping gene GAPDH. The primer sequences were as follows: human GAPDH, forward primer 5′-AGGTGAAGGTCGGAGTCA-3′, reverse primer 5′-GGTCATTGATGGCAACAA-3′; human IL-8, forward primer 5′-TTTTGCCAAGGAGTGCTAAAGA-3′, reverse primer 5′-AACCCTCTGCACCCAGTTTTC-3′; human IL-6, forward primer 5′ ACTCACCTCTTCAGAACGAATTG-3′, reverse primer 5′-CCATCTTTGGAAGGTTCAGGTTG-3′; human CCND1, forward primer 5′-GCTGCGAAGTGGAAACCATC-3′, reverse primer 5′-CCTCCTTCTGCACACATTTGAA-3′; human COX2, forward primer 5′-CTGGCGCTCAGCCATACAG-3′, reverse primer 5′-CGCACTTATACTGGTCAAATCCC-3′.

### 4.9. Plasmid Transfection

HEK293 or A549 cells were transfected using Lipofectamine 3000 or Lipofectamine LTX Transfection Reagent (Life Technologies, # L3000015) according to the manufacturer’s protocol.

### 4.10. Knockout by CRISPR/Cas9

The single guide RNA (sgRNA) targeting sequences were as follows: HOIP, 5′-ATGCAAGTTCTCGTACGCCC-3′; PKP2, 5′-GCGAGCAGTGAGTATGCTCG-3′. The sgRNA was cloned into lentiCRISPR v2 vector (53) (Addgene, Watertown, MA, USA). The lentiviral construct was transfected with psPAX2 and pMD2G into HEK293T cells using PEI. After 48 h, the media containing lentivirus were collected. The targeted cells were infected with the media containing the lentivirus supplemented with 10 μg/mL polybrene. Cells were selected with 10 μg/mL puromycin for 14 days. Single clones were expanded for knockout confirmation by Western blotting.

### 4.11. Statistical Analysis

The sample size was sufficient for data analyses. Data were statistically analyzed using the software GraphPad Prism 9 (San Diego, CA, USA). Significant differences between the indicated pairs of conditions are shown by asterisks in the figures of this paper (* *p* < 0.05; ** *p* < 0.01; *** *p* < 0.001; **** *p* < 0.0001).

## 5. Conclusions

In this study, we first demonstrate that PKP2 interacts with the kinase domain of EGFR, and PKP2 is required for EGFR-mediated NF-κB signaling and NF-κB-regulated gene expression. Secondly, the N-terminal domain PKP2 interacts with the NZFs of HOIP and PKP2 promotes NF-κB activity via HOIP. Thirdly, upon EGF stimulation, EGFR recruits HOIP through PKP2, thereby activating linear ubiquitination of NEMO. Fourthly, HOIP deficiency and the HOIP inhibitor HOIPIN-8 inhibit A431 tumor cell proliferation and tumor development in the xenograft mouse model. Lastly, HOIPIN-8 suppresses breast cancer cell proliferation and clonogenicity by blocking EGFR-mediated NF-κB activation. Overall, our study defines a novel EGFR-NF-κB signaling axis through PKP2 and HOIP, and perturbation of HOIP E3 ubiquitin ligase activity is a potential therapeutic for tumors driven by EGFR activation.

## Figures and Tables

**Figure 1 ijms-22-11875-f001:**
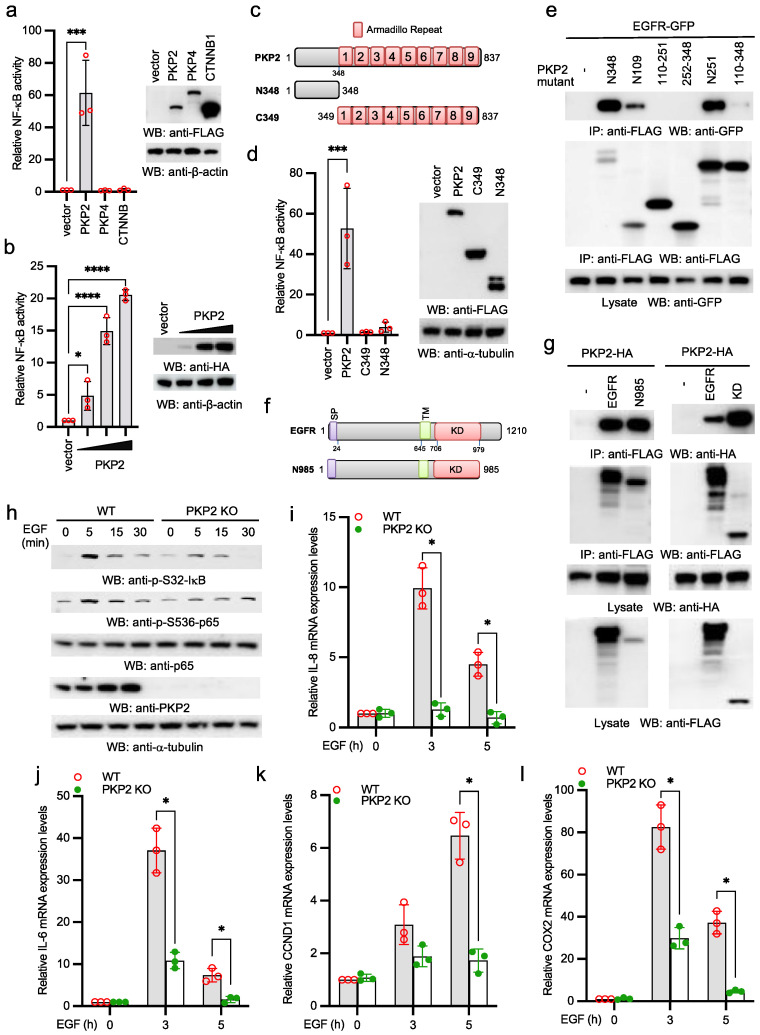
PKP2 is required for EGFR-induced NF-κB signaling. (**a**) FLAG-tagged PKP2, PKP4, or CTNNB1 was transfected with pRL-SV40 (Renilla luciferase as an internal control) and NF-κB-Luc into HEK293 cells. After 48 h, cells were collected, and the ratio of firefly luciferase to Renilla luciferase was calculated to determine the relative reporter activity. Right panel shows protein expression levels by Western blotting. (**b**) Different doses of PKP2 were transfected with pRL-SV40 and NF-κB-Luc into HEK293 cells. After 48 h, cells were collected, and the ratio of firefly luciferase to Renilla luciferase was calculated to determine the relative reporter activity. Right panel shows protein expression levels by western blotting. (**c**) Schematics of PKP2 mutants. (**d**) PKP2 or the indicated mutant was transfected with pRL-SV40 and NF-κB-Luc into HEK293 cells. After 48 h, cells were collected, and the ratio of firefly luciferase to Renilla luciferase was calculated to determine the relative reporter activity. Right panel shows protein expression levels by Western blotting. (**e**) GFP-tagged EGFR (EGFR-GFP) was transfected with the indicated FLAG-tagged PKP2 mutants into HEK293 cells. After 48 h, cell lysates were collected and then immunoprecipitated and blotted as indicated. (**f**) Schematics of EGFR mutants. SP: signal peptide; TM: transmembrane; KD: kinase domain. (**g**) HA-tagged PKP2 (PKP2-HA) was transfected with the indicated FLAG-tagged EGFR or the indicated mutants into HEK293 cells. After 48 h, cell lysates were collected and then immunoprecipitated and blotted as indicated. (**h**) PKP2 wild type and knockout A431 cells were stimulated with 5 ng/mL EGF for indicated times. Cell lysates were collected and blotted as indicated. (**i**–**l**) PKP2 wild type and knockout A431 cells were stimulated with 5 ng/mL EGF for indicated times. Real-time PCR was performed to determine the relative mRNA levels of IL-8 (**i**), IL-6 (**j**), CCND1 (**k**), and COX2 (**l**). All experiments were biologically repeated three times. (**a**,**b**,**d**,**i**–**l)** Data represent means ± s.d. of three independent experiments. The *p*-value was calculated by one-way ANOVA followed by Dunnett’s multiple comparisons test (**a**,**b**,**d**) or two-way ANOVA followed by Sidak’s multiple comparisons test (**i**–**l**) (* *p* < 0.05, *** *p* < 0.001, **** *p* < 0.0001).

**Figure 2 ijms-22-11875-f002:**
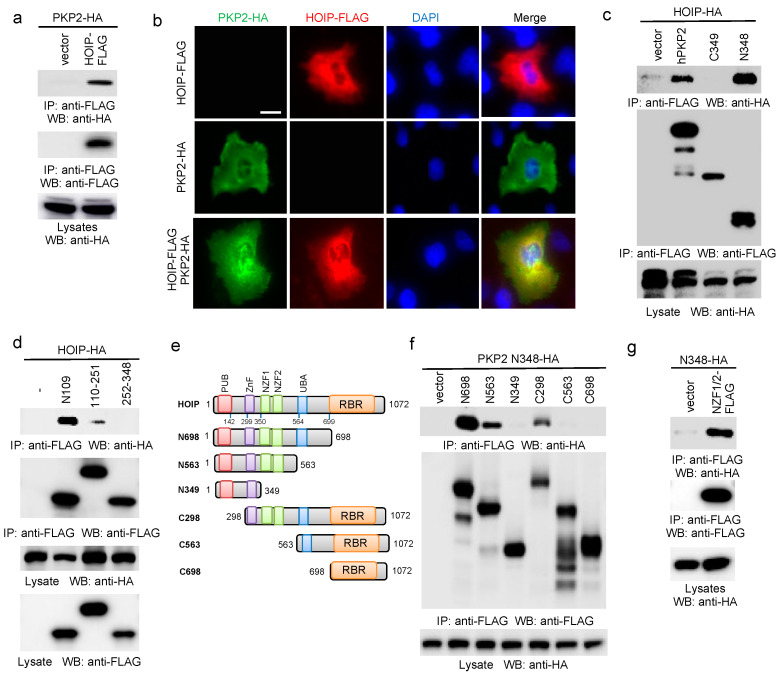
PKP2 interacts and co-localizes with HOIP. (**a**) FLAG-tagged HOIP (HOIP-FLAG) was co-transfected with PKP2-HA into HEK293 cells. After 48 h, cell lysates were immunoprecipitated with an anti-FLAG antibody and blotted as indicated. (**b**) HOIP-FLAG and PKP2-HA were transfected into A431 cells. After 48 h, cells were fixed and stained as indicated. FLAG: red; HA: green; DAPI, blue. Bar = 10 μm. (**c**) HA-tagged HOIP (HOIP-HA) was transfected with PKP2-FLAG or the indicated mutants into HEK293 cells. After 48 h, cell lysates were collected and then immunoprecipitated and blotted as indicated. (**d**) HOIP-HA was transfected with the indicated FLAG-tagged PKP2 mutants into HEK293 cells. After 48 h, cell lysates were collected and then immunoprecipitated and blotted as indicated. (**e**) Schematics of HOIP mutants. PUB: PNGase/UBA- or UBX-containing domain; ZnF: zinc finger; NZF: Npl4-type zinc finger; UBA: ubiquitin-associated domain; RBR: RING-IBR-RING domain. (**f**) HA-tagged PKP2 N348 was transfected with the indicated FLAG-tagged HOIP mutants into HEK293 cells. After 48 h, cell lysates were collected and then immunoprecipitated and blotted as indicated. (**g**) HA-tagged PKP2 N348 was transfected with FLAG-tagged two NZF domains (NZF1/2) into HEK293 cells. After 48 h, cell lysates were collected and then immunoprecipitated and blotted as indicated.

**Figure 3 ijms-22-11875-f003:**
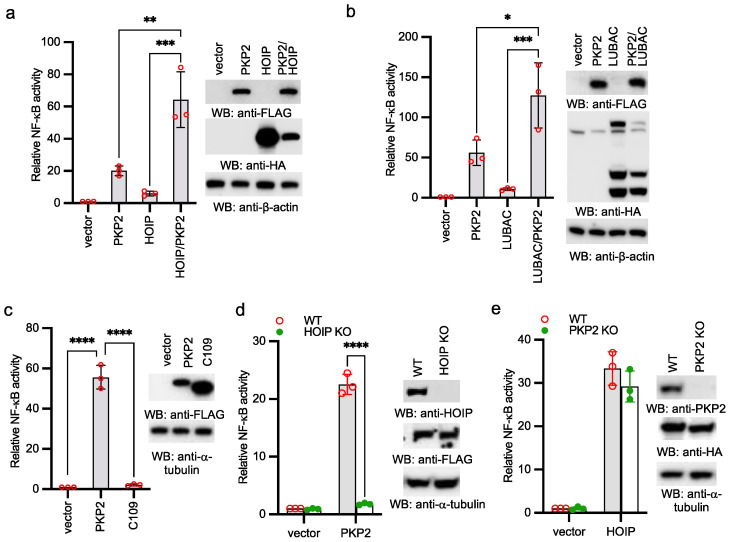
PKP2 activates NF-κB via HOIP. (**a**) PKP2 was co-transfected with vector or HOIP, together with pRL-SV40 and NF-κB-Luc, into HEK293 cells. After 48 h, cells were collected, and the ratio of firefly luciferase to Renilla luciferase was calculated to determine the relative reporter activity. Right panel shows protein expression levels by Western blotting. (**b**) PKP2 was co-transfected with vector or LUBAC (HOIP, HOIL-1, and SHARPIN), together with pRL-SV40 and NF-κB-Luc, into HEK293 cells. After 48 h, cells were collected, and the ratio of firefly luciferase to Renilla luciferase was calculated to determine the relative reporter activity. Right panel shows protein expression levels by Western blotting. (**c**) PKP2 or the C109 (aa 109-837) was co-transfected with pRL-SV40 and NF-κB-Luc into HEK293 cells. After 48 h, cells were collected, and the ratio of firefly luciferase to Renilla luciferase was calculated to determine the relative reporter activity. Right panel shows protein expression levels by Western blotting. (**d**) HOIP wild type (WT) and knockout (KO) HEK293 cells were transfected with vector or PKP2-FLAG, together with pRL-SV40 and NF-κB-Luc. After 48 h, cells were collected, and the ratio of firefly luciferase to Renilla luciferase was calculated to determine the relative reporter activity. Right panel shows protein expression levels by Western blotting. (**e**) PKP2 WT and KO HEK293 cells were transfected with vector or HOIP-FLAG, together with pRL-SV40 and NF-κB-Luc. After 48 h, cells were collected, and the ratio of firefly luciferase to Renilla luciferase was calculated to determine the relative reporter activity. Right panel shows protein expression levels by western blotting. (**a**–**e**) All experiments were biologically repeated three times. Data represent means ± s.d. of three independent experiments. The *p*-value was calculated by one-way ANOVA (**a**–**c**), or two-way ANOVA followed by Sidak’s multiple comparisons test (**d**,**e**) (* *p* < 0.05, ** *p* < 0.01, *** *p* < 0.001, **** *p* < 0.0001).

**Figure 4 ijms-22-11875-f004:**
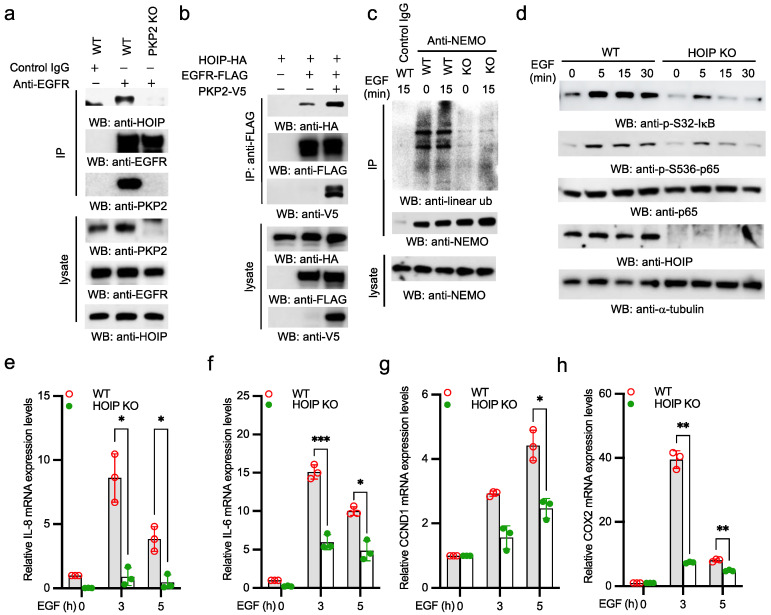
EFGR recruits HOIP and induces linear polyubiquitin-mediated NF-κB activation. (**a**) PKP2 WT and KO A431 cells were stimulated with 5 ng/mL EGF for 30 min. Cell lysates were immunoprecipitated with an anti-EGFR antibody or control IgG and blotted as indicated. (**b**) EGFR, HOIP, and PKP2 were transfected in the indicated combination into HEK293 cells. After 48 h, cell lysates were immunoprecipitated with an anti-FLAG antibody and blotted as indicated. (**c**) HOIP WT and KO A431 cells were stimulated with 5 ng/mL EGF for designated times. Cell lysates were immunoprecipitated with an anti-NEMO antibody or control IgG and blotted as indicated. (**d**) HOIP WT and KO A431 cells were stimulated with 5 ng/mL EGF for designated times. Cell lysates were blotted as indicated. (**e**–**h**) HOIP WT and KO A431 cells were stimulated with 5 ng/mL EGF for indicated times. Real-time PCR was performed to determine the relative mRNA levels of IL-8 (**e**), IL-6 (**f**), CCND1 (**g**), and COX2 (**h**). All experiments were biologically repeated three times. Data represent means ± s.d. of three independent experiments. The *p*-value was calculated by two-way ANOVA followed by Sidak’s multiple comparisons test (* *p* < 0.05, ** *p* < 0.01, *** *p* < 0.001).

**Figure 5 ijms-22-11875-f005:**
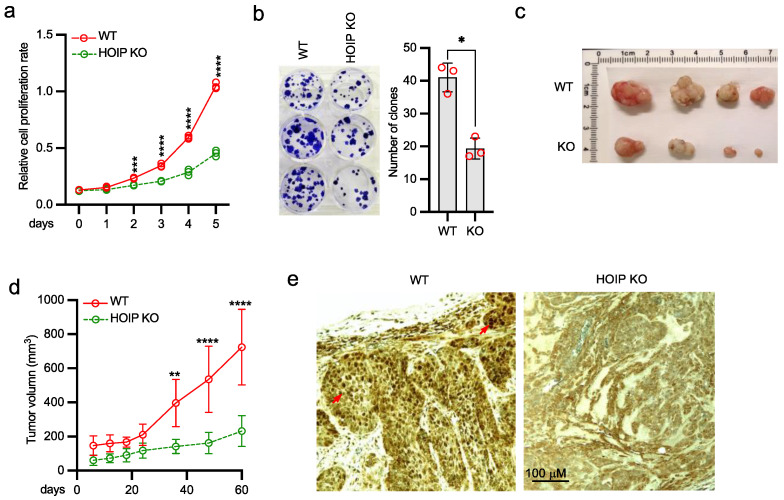
HOIP deficiency inhibits tumor cell proliferation and tumor development. (**a**) The same number of HOIP WT and KO A431 cells were cultured at day 0. Cells were collected at the indicated times for MTT assays to determine cell growth rate. All experiments were biologically repeated three times. Data represent means ± s.d. of three independent experiments. The *p*-value was calculated by two-way ANOVA followed by Sidak’s multiple comparisons test (*** *p* < 0.001, **** *p* < 0.0001). (**b**) HOIP WT and KO A431 cells were cultured to form colonies. All experiments were biologically repeated three times. Data represent means ± s.d. of three independent experiments. The *p* value was calculated (two-tailed Student’s t-test) by comparison with wild type cells (* *p* < 0.05). (**c**) Representative figure of tumors from nude mice xenografted with HOIP WT or KO A431 cells. (**d**) Growth rate of tumors from nude mice xenografted with HOIP WT or KO A431 cells. The *p*-value was calculated by two-way ANOVA followed by Sidak’s multiple comparisons test (** *p* < 0.01, **** *p* < 0.0001). (**e**) Representative IHC images of Ki-67 staining with hematoxylin counterstain in HOIP WT and KO tumors. Arrows indicate the high intensity staining of Ki-67 in the nuclei.

**Figure 6 ijms-22-11875-f006:**
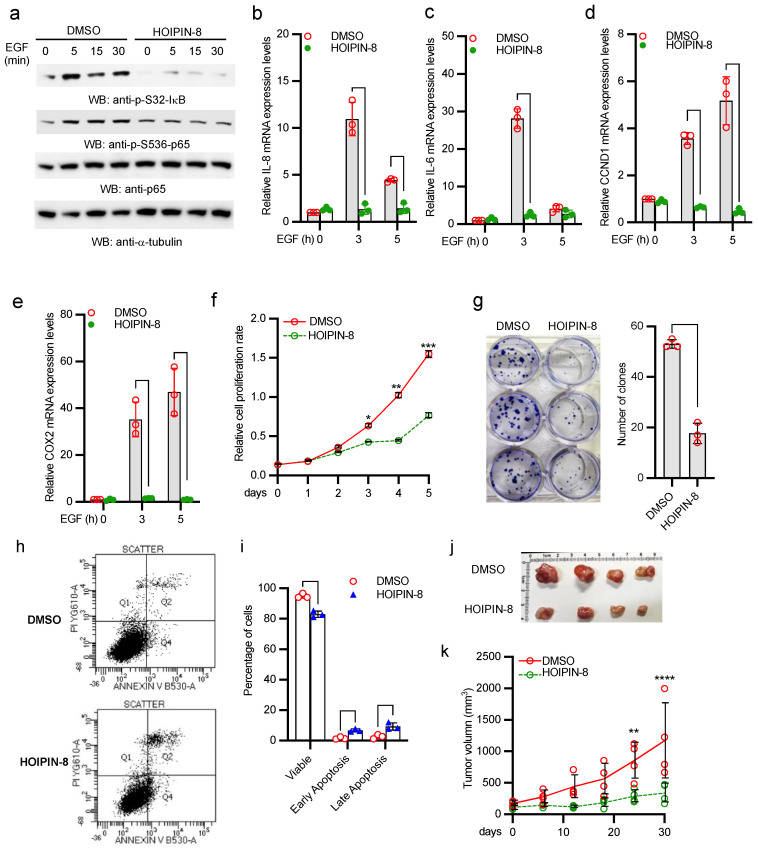
HOIPIN-8 inhibits EGFR-mediated NF-κB activation and tumor development. (**a**) A431 cells were treated with DMSO or 30 μM HOIPIN-8 for 2 h. Then, cells were stimulated with 5 ng/mL EGF for designated times. Cell lysates were blotted as indicated. (**b**–**e**) A431 cells were treated with DMSO or 30 μM HOIPIN-8 for 2 h. Then, cells were stimulated with 5 ng/mL EGF for designated times. Real-time PCR was performed to determine the relative mRNA levels of IL-8 (**b**), IL-6 (**c**), CCND1 (**d**), and COX2 (**e**). All experiments were biologically repeated three times. Data represent means ± s.d. of three independent experiments. The *p*-value was calculated by two-way ANOVA followed by Sidak’s multiple comparisons test (* *p* < 0.05, ** *p* < 0.01). (**f**) A431 cells were treated with DMSO or 30 μM HOIPIN-8. Cells were collected at the indicated times for MTT assays. All experiments were biologically repeated three times. Data represent means ± s.d. of three independent experiments. The *p*-value was calculated by two-way ANOVA followed by Sidak’s multiple comparisons test (* *p* < 0.05, ** *p* < 0.01, *** *p* < 0.001). (**g**) A431 cells were treated with DMSO or 30 μM HOIPIN-8 for 72 h. Then, cells were cultured for additional 7 days and then colonies were photographed. All experiments were biologically repeated three times. Data represent means ± s.d. of three independent experiments. The *p* value was calculated (two-tailed Student’s *t*-test) by comparison with the DMSO group (** *p* < 0.01). (**h**,**i**) A431 cells were treated with DMSO or 60 µM of HOIPIN-8 for 48 h. Then, apoptosis assays were performed using flow cytometry after staining with annexin V-FITC/propidium iodide (PI). (**h**) Representative scatter plots of PI (y-axis) vs. annexin V (x-axis). (**i**) Percentage of viable, early apoptotic, and late apoptotic cells. Data represent means ± s.d. of three independent experiments. The *p*-value was calculated by two-way ANOVA followed by Sidak’s multiple comparisons test (** *p* < 0.01, *** *p* < 0.001, **** *p* < 0.0001). (**j**) Representative figure of tumors from A431 cells xenograft nude mice treated with DMSO or HOIPIN-8. (**k**) Growth rate of tumors from A431 cells xenograft nude mice treated with DMSO or HOIPIN-8. The *p*-value was calculated by two-way ANOVA followed by Sidak’s multiple comparisons test (** *p* < 0.01, **** *p* < 0.0001).

**Figure 7 ijms-22-11875-f007:**
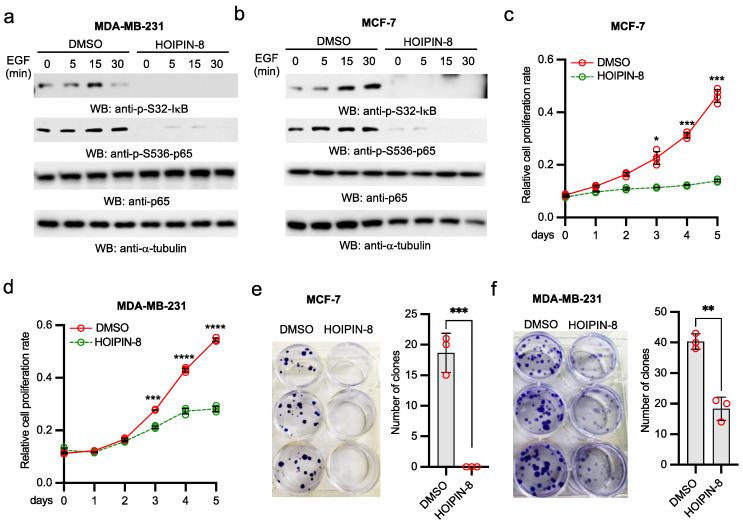
HOIPIN-8 impairs EGFR-mediated NF-κB activation in breast cancer cells and inhibits their proliferation and clonogenicity. (**a**) MDA-MB-231 cells were treated with DMSO or 30 μM HOIPIN-8 for 2 h. Then, cells were stimulated with 5 ng/mL EGF for designated times. Cell lysates were blotted as indicated. (**b**) MCF-7 cells were treated with DMSO or 30 μM HOIPIN-8 for 2 h. Then, cells were stimulated with 5 ng/mL EGF for designated times. Cell lysates were blotted as indicated. (**c**,**d**) MCF-7 (**c**) and MDA-MB-231 (**d**) cells treated with DMSO or 30 μM HOIPIN-8. Cells were collected at the indicated times for MTT assays. All experiments were biologically repeated three times. Data represent means ± s.d. of three independent experiments. The *p*-value was calculated by two-way ANOVA followed by Sidak’s multiple comparisons test (* *p* < 0.05, *** *p* < 0.001, **** *p* < 0.0001). (**e**–**f**) MCF-7 (**e**) and MDA-MB-231 (**f**) cells were treated with DMSO or 30 μM HOIPIN-8 for 72 h. Then, cells were cultured for additional 7 days and then colonies were photographed. All experiments were biologically repeated three times. Data represent means ± s.d. of three independent experiments. The *p* value was calculated (two-tailed Student’s *t*-test) by comparison with the DMSO group (** *p* < 0.01, *** *p* < 0.001).

**Table 1 ijms-22-11875-t001:** PCR primers used for cloning.

Construct Name	Forward Primer	Reverse Primer	Method
PKP2 N109	gtcataggttttaggaacaggggaacg	gaggattacaaggatgacgacg	Mutagenesis
PKP2 N251	gctgcggctggtccctggcctgg	gaggattacaaggatgacgacg	Mutagenesis
PKP2 110-251	gcactcgagatgctaaaggctggcacaactgc	gtaaagcttgctgcggctggtccctgg	Cloning: XhoI, HindIII
PKP2 252-348	gcactcgagatgggcaacctcttggagaaggag	gtaaagcttgtctgcattccccagctgggag	Cloning: XhoI, HindIII
PKP2 110-348	gcactcgagatgctaaaggctggcacaactgc	gtaaagcttgtctgcattccccagctgggag	Cloning: XhoI, HindIII
EGFR N985	aaggtagcgctgggggtctc	gaggattacaaggatgacgacg	Mutagenesis
EGFR KD	gcactcgaggccaccatgatcttgaaggaaactgaattc	gtagcggccgcttcatccccctgaatgacaaggtagc	Cloning: XhoI, NotI
HOIP N698	gcagcggccgcatgccgggggaggaagag	gtaatcgatctcctgggcaagcaagcg	Cloning: ClaI, NotI
HOIP N563	gcagcggccgcatgccgggggaggaagag	gtaatcgatgccatgacgatccagccaggctc	Cloning: ClaI, NotI
HOIP N349	gcagcggccgcatgccgggggaggaagag	gtaatcgataagatcaggttctaggcctccag	Cloning: ClaI, NotI
HOIP C298	gcagcggccgcatggcaagtgctcatttgccctggcac	gtaatcgatcttccgcctgcgggggatac	Cloning: ClaI, NotI
HOIP C563	gcagcggccgcatgggcaaccttgatgaagctgtggag	gtaatcgatcttccgcctgcgggggatac	Cloning: ClaI, NotI
HOIP C698	gcagcggccgcatggagtgtgccgtgtgtggctgg	gtaatcgatcttccgcctgcgggggatac	Cloning: ClaI, NotI
HOIP NZF1/2	gcagcggccgcatgggaggcctagaacctgatc	gtaatcgatggcatgttgtgctggaatgg	Cloning: ClaI, NotI

## Data Availability

The authors declare that the data supporting the findings of this study are available within the article and from the corresponding author upon request.

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
