# Peer review of "Linear Ubiquitination Mediates EGFR-Induced NF-κB Pathway and Tumor Development"

_ijms, 2021, doi:10.3390/ijms222111875_

Round 1
Reviewer 1 Report
In the manuscript entitled “Linear ubiquitination mediates EGFR-induced NF-kB pathway 2 and tumor development” the authors describe that PKP2 connects EGF signalling with the NF-kB pathway through a mechanism that involves LUBAC-mediated ubiquitination of NEMO. They also show evidence of the potential use of HOIPIN-8 to target this signalling pathway and the in vivo effect of targeting HOIP in a xenograft model. Overall, the work is very well written and organised, accompanied with data of very good quality to support the results discussed.
Major comments:
- The authors claim that targeting HOIP either by knocking it out or by using the inhibitor HOIPIN-8 decreases the proliferation ability of the cells as well as their clonogenicity and reduces tumour size. However, it would be very interesting to see if that effect is due to an increase in cell death/apoptosis, since targeting HOIP would not only be beneficial as a strategy to promote tumour regression but also to sensitize cells to apoptosis as already suggested.
- In the last section in which the authors show the effect of HOIPIN-8 in two different breast cancer cell lines, it would be interesting to see whether the effect of targeting HOIP over the NF-kB is dependent on PKP2.
Minor comments:
- The authors suggest that the mechanism described is dependent on NEMO ubiquitination. Giving evidence of this mechanism by knocking down NEMO levels would enhance this hypothesis.
- Line 318 – there is an unfinished sentence.
Reviewer 2 Report
In their manuscript “ Linear ubiquitination mediates EGFR-induced NF-kB pathway 2 and tumor development " the Fang Hua et al. intended to explore the prospective treatment strategies for targeted cancer therapy which underlying the mechanism of disruption of HOIP E3 ubiquitin ligase activity and a novel linear ubiquitination signaling axis of EGFR induced NF-kB pathway to promote tumor development.
This constitutes a large and broadly coherent body of work and their findings are:
- The involvement of PKP2 and its both domains for NF-kB activation. The significance of PKP2 for EGF-induced NF-kB activation and NF-kB regulated gene expression.
- HOIP (NZFs) are essential for the binding to the N-terminus of PKP2 and PKP2 promotes NF-kB activation via HOIP.
- EGFR activates the linear ubiquitination of NEMO and NF-kB transcriptional activity via HOIP.
- Silencing of HOIP (genetic mutation/inhibitor) inhibits A431 tumor cell proliferation and tumor development.
- HOIP inhibitor also inhibits EGFR-mediated NF-kB activation in breast cancer cells and suppresses breast cancer cell proliferation and clonogenicity.
However, there are a number of substantial issues that need to be resolved, listed below.
- The whole study is performed with well study design. However, I was struggling with the arrangement of information in the introduction section which was presented mostly without giving rationales/hypotheses, which leaves the reader wondering why the authors picked randomly those proteins to study in a particular portion. Only in the result and discussion part, this becomes a bit clearer. I recommend reconsidering the order of information or the introduction part can be made simpler/elucidated.
- The conclusion portion is too short to explain the whole study which is diluting the importance of the study.
- Please mention elaborately the processing of Point mutations and deletions of HOIP, EGFR, and PKP2 in the materials and method section.
- In Fig.5 the authors stated that “perturbation of linear ubiquitin-mediated NF-kB activation inhibits A431 tumor cell proliferation and tumor development” but they have not performed an assay to measure NF-kB activation or NF-kB regulated gene expression.
Author Response
The whole study is performed with well study design. However, I was struggling with the arrangement of information in the introduction section which was presented mostly without giving rationales/hypotheses, which leaves the reader wondering why the authors picked randomly those proteins to study in a particular portion. Only in the result and discussion part, this becomes a bit clearer. I recommend reconsidering the order of information or the introduction part can be made simpler/elucidated.
We thank the reviewer for the constructive suggestion. We now add the rationale for this study and the central hypothesis in the Introduction section (lines 67-70).
The conclusion portion is too short to explain the whole study which is diluting the importance of the study.
We now expand the conclusion (lines 536-544).
Please mention elaborately the processing of Point mutations and deletions of HOIP, EGFR, and PKP2 in the materials and method section.
A detailed method was added. In addition, we added Table 1 for primer sequences (lines 425-434).
In Fig.5 the authors stated that “perturbation of linear ubiquitin-mediated NF-kB activation inhibits A431 tumor cell proliferation and tumor development” but they have not performed an assay to measure NF-kB activation or NF-kB regulated gene expression.
Corrected (line 243).